# Forward Osmosis Technology and Its Application on Microbial Fuel Cells: A Review

**DOI:** 10.3390/membranes12121254

**Published:** 2022-12-12

**Authors:** Yang Zhao, Liang Duan, Xiang Liu, Yonghui Song

**Affiliations:** 1State Key Joint Laboratory of Environment Simulation and Pollution Control, School of Environment, Tsinghua University, Beijing 100084, China; 2Chinese Research Academy of Environmental Sciences, Beijing 100012, China

**Keywords:** forward osmosis, bioelectrochemical systems, energy recovery, potential applications

## Abstract

As a new membrane technology, forward osmosis (FO) has aroused more and more interest in the field of wastewater treatment and recovery in recent years. Due to the driving force of osmotic pressure rather than hydraulic pressure, FO is considered as a low pollution process, thus saving costs and energy. In addition, due to the high rejection rate of FO membrane to various pollutants, it can obtain higher quality pure water. Recovering valuable resources from wastewater will transform wastewater management from a treatment focused to sustainability focused strategy, creating the need for new technology development. An innovative treatment concept which is based on cooperation between bioelectrochemical systems and forward osmosis has been introduced and studied in the past few years. Bioelectrochemical systems can provide draw solute, perform pre-treatment, or reduce reverse salt flux to help with FO operation; while FO can achieve water recovery, enhance current generation, and supply energy sources for the operation of bioelectrochemical systems. This paper reviews the past research, describes the principle, development history, as well as quantitative analysis, and discusses the prospects of OsMFC technology, focusing on the recovery of resources from wastewater, especially the research progress and existing problems of forward osmosis technology and microbial fuel cell coupling technology. Moreover, the future development trends of this technology were prospected, so as to promote the application of forward osmosis technology in sewage treatment and resource synchronous recovery

## 1. Introduction to Forward Osmosis (FO)

### 1.1. Principle of Forward Osmosis

Forward osmosis is a separation process that uses the osmotic pressure difference between feed solution (FS) and draw solution (DS) on both sides of the forward osmosis membrane as the driving force without external pressure to make water flow spontaneously from the feed solution (low osmotic pressure) to the drawn solution (high osmotic pressure) as shown in Figure 1. In this process, the FO membrane selectively penetrates water molecules to intercept and remove pollutants and ions in water. Forward osmosis (FO) is based on the natural phenomena of osmotic processes and can extract clean water from wastewater [1].

Compared with other membrane systems, FO has many advantages, such as high energy efficiency, high salt discharge rate, low membrane pollution, and low salt water discharge. Therefore, once FO becomes advantageous, determine the appropriate DS based on different wastewater types. The pore diameter of the second FO membrane is only 0.3–0.5 nm, allowing a high solute rejection rate, making it an ideal choice for desalination, removal of heavy metals, and removal of micro pollutants, such as cell inhibitory drugs and endocrine disruptors. In addition, FO does not require pre-treatment of wastewater. Another key advantage of the FO process is its low pollution tendency [2]. Reversible fouling is the most common type of membrane fouling and can be repaired using simple hydraulic cleaning. Seawater is one of the most commonly used DS, which has been diluted and can be safely discharged back to the sea without any treatment. The concentrated FS can be produced by anaerobic digestion.

### 1.2. Development of Forward Osmosis(FO)

The development of FO technology and membrane materials has mainly gone through three stages:

Stage 1: A new process for desalination of seawater based on the principle of FO membrane was proposed for the first time. However, at this stage, a special FO membrane was not developed, but the reverse osmosis membrane was used for FO research. Due to the dense support layer of the reverse osmosis membrane, serious internal concentration polarization was caused when it was applied to FO, resulting in low FO performance [3].

Stage 2: Started to explore a semi permeable membrane more suitable for the FO process. HTI Company of the United States used polyester mesh to replace the RO membrane support layer, developed an asymmetric cellulose triacetate FO membrane (CTA membrane) with better performance, and realized commercial application in the field survival water purification equipment and food concentration. However, compared with the RO process, FO water flux is still at a low level. Moreover, the mass transfer mechanism of FO process and the study of extraction solution are still not in-depth.

Stage 3: Further development has been made in exploring FO mass transfer mechanism and model, developing efficient extraction solutions, and developing high-performance FO membranes. The researchers successfully prepared polyamide composite membrane (TFC) through interfacial polymerization, which improved the water flux and salt rejection of the FO process, and had a wider pH application range than CTA membrane. Different from the traditional membrane separation process, the forward osmosis process uses the osmotic pressure difference between two solutions to drive water through the semi permeable membrane, without additional pressure, so it has the advantage of low energy consumption [4].

The biggest feature of FO technology is osmotic pressure driving, which is essentially different from other membrane separation processes. Therefore, compared with traditional pressure driven membrane separation processes (such as reverse osmosis and nanofiltration), FO technology has the following advantages.

◾
**Low energy consumption**


No hydraulic pressure is required in the operation process, so the FO process has the advantage of low energy consumption, especially in applications where the extract does not need to be recycled, such as the diluted fertilizer extract, directly used for agricultural irrigation, and the diluted seawater extract, directly discharged, which can obviously reflect the low energy consumption advantage of FO process [5].

◾
**Light membrane pollution and high reversibility**


No hydraulic pressure can prevent pollutants on the membrane surface from being compacted, resulting in light FO membrane pollution and high reversibility.

◾
**High pollutant retention rate and good effluent quality**


The pore diameter of FO membrane is very small (about 0.25–0.3 nm), which has an excellent removal effect on ions and micro pollutants in water. Therefore, FO technology with low energy consumption, low pollution, and high retention has a very broad application prospect [6].

### 1.3. Concentration Polarization

Concentration polarization is a common phenomenon in all membrane separation processes, and the forward osmosis process is no exception. Concentration polarization is due to the fact that during the membrane separation process of water and solute, the solute of the feed solution accumulates on the membrane surface layer, and one side of the draw solution is diluted by water, resulting in the phenomenon that the effective osmotic pressure of the membrane layer is far less than the osmotic pressure difference of the solution itself on both sides [7]. Concentration polarization not only reduces osmotic driving force, thereby reducing water flux and increasing solute diffusion, but also aggravates membrane pollution. Due to the asymmetric structure of the forward osmosis membrane, external concentration polarization and internal concentration polarization are prone to occur. The outer concentration polarization occurs on the membrane surface and can be reduced or eliminated by hydraulic conditions [8]. The inner concentration polarization occurs in the support layer of the membrane, which seriously affects the performance of the forward osmosis membrane. In the process of forward osmosis, there are two commonly used operation modes:

**FO mode or AL-FS mode.** The active layer of the feed solution towards the membrane.

**PRO mode or AL-DS mode.** The active layer of the absorption solution towards the membrane.

Different membrane orientation will lead to different dilution or concentration polarization. Figure 2 describes the concentration polarization diagram of FO and PRO modes [9].

In the AL-FS mode, the water molecules of the feed solution enter the absorption solution side through the membrane, while the solute gradually accumulates in the active layer of the membrane, making the concentration of the solute on the membrane surface greater than its concentration in the solution, forming a concentrated external concentration polarization. At the same time, the water permeates the active layer with gradually diluting the extract of the support layer and then the diluted internal concentration polarization occurs. In AL-DS mode, the solute in the feed solution gradually accumulates in the membrane support layer and the concentrated inner concentration polarization occurs [10]. The absorption solution near the active layer is diluted by the transferred water, which reduces the concentration and polarizes the diluted external concentration difference. Therefore, regardless of the membrane orientation, the concentration polarization will reduce the osmotic pressure, resulting in a decrease in water flux. In the process of forward osmosis, the internal concentration polarization occurs in the support layer and cannot be removed through optimization of hydraulic conditions, which is the main reason for the decline of water flux [11].

### 1.4. Membrane Fouling

Membrane fouling involves solutes and/or particles on the membrane surface and in the membrane hole or the feed spacer is blocked. This may cause dirt, scaling, or damage of the membrane. The main pollutants in natural and damaged water bodies are microorganisms, organic substances, and inorganic substances (scaling). When wastewater is used, due to the existence of microorganisms and the secretion of extracellular polymeric substances (EPS) to establish biofilm integrity, biological scaling may be the most limiting factor [12]. Biological scaling is affected by influent water quality, membrane physical and chemical properties and operating conditions. In a FO-MBR study, biological deposition had little effect on water permeability, but the mass transfer coefficient was seriously reduced and ICP was enhanced. In seawater FO, silica scaling or membrane biological scaling may occur through transparent outer polymer particles (TEP). Organic pollution varies depending on the water supply used. The wastewater consists of mobile organic matter (EfOM), including soluble microbial products and natural organic matter (NOM). NOM has been found to be a serious pollutant in many membrane processes, including FO [13]. Therefore, it is important to simulate the behavior of these complex feeds to include all or the most important dirt. Model fouling, using, e.g., sodium alginate or alginate, bovine serum albumin (BSA), and Aldrich humic acid (AHA), has been used to test the severity of NOM fouling on FO membrane. Alginic acid is related to the hydrophilic part of EfOM, AHA represents humic acid, and BSA represents protein part [14].

Immediate fouling detection ensures and restores membrane performance. Determining the scaling potential of the feed can help predict scaling, However, once fouling occurs on the membrane surface, off-line methods may be required for future preventive measures. Non invasive visual online methods can detect early signs of fouling in real time, such as flow decline, solute rejection, and NPD change operating parameters (temperature, feed TDS, penetrant flow, recovery). Figure 3 summarizes the fouling detection technology in which feed and FO membrane contamination are involved [15].

### 1.5. Application of FO Technology

The idea of wastewater treatment has changed from the original “pollutant removal up to standard discharge” to the idea of “resource and energy recycling”, which can realize water resource regeneration, energy production, and value-added product output [16].The advantages of FO technology, such as low energy consumption, light pollution and high interception rate, make it widely used and researched in wastewater treatment, specifically including water resource regeneration and nitrogen and phosphorus nutrient recovery [17].

◾
**Water resources regeneration**


Due to the high interception of FO membrane, most of the pollutants in wastewater can be removed, and high-quality effluent can be obtained to realize the regeneration and reuse of water resources. The FO wastewater treatment and resource recovery unit is composed of two parts, namely the FO treatment system and extraction liquid recovery water purification system. Zhang et al. studied the treatment effect of FO membrane on the effluent of the secondary sedimentation tank and used solar radiation to drive electrodialysis to recover the diluted extract, which can meet the drinking water standard [18].

◾
**Recovery of nitrogen and phosphorus nutrients**


Wastewater contains rich nutrients, such as nitrogen and phosphorus. If discharged directly, it will not only reduce the effluent quality, but also cause eutrophication of the water body. Recycling nitrogen, phosphorus, and other nutrients as fertilizers is an urgent need for sustainable development of wastewater treatment.

The dense membrane pore of FO membrane can effectively intercept and concentrate ammonia, nitrogen, and phosphate in wastewater for subsequent crystallization and recovery [13]. At present, it is successfully used for concentration and recovery of nitrogen and phosphorus resources in anaerobic digestion liquid and urine shown in Figure 4. In addition, using the reverse diffusion characteristics of the FO draw solution, with the salt solution of magnesium bivalent as the extracting solution, nitrogen and phosphorus in the synthetic urine are recovered by FO technology [19]. After FO treatment, magnesium ions entering the concentrated solution form struvite precipitation with phosphorus. The diluted extracting solution of recovered urea is used for the direct irrigation of green walls, parks, or urban agriculture.

## 2. FO and Bioelectrochemical System Technology (BES)

### 2.1. Coupling Advantages of Forward Osmosis Technology and Microbial Fuel Cell Technology

The forward osmosis microbial fuel cell technology, which combines the advantages of forward osmosis technology and microbial fuel cell technology, improves the power generation performance of MFCs and demonstrates good performance in water recovery. Coupled technology links BES and FO units externally through a hydraulic connection [20].

Wastewater could be used as a source of fuel for BES, with the benefit of accomplishing wastewater treatment. In recent years, BES has been researched in the context of treating wastewater and extracting the waste energy extensively, with the representing technology, microbial fuel cell (MFC). For example, MFCs may produce up to 1.43 kWh m^−3^ from a primary sludge or 1.8 kWh m^−3^ from a treated effluent [21]. Theoretically, BES can convert maximum 100% of chemical energy into electricity.

However, there is always some energy lost through (1) coulombic loss where organics are not converted to electrical current at 100%, and (2) electrochemical potential or voltage loss. Nevertheless, the reported energy conversion efficiency for MFC can reach 80% which is much higher than 33% for typical heat engine combustion of methane gas. An example of a coupled technology is to connect an MEC to an FO unit for recovering ammonium from a synthetic wastewater and then applying the recovered ammonium as a draw in the subsequent FO process [22].

In an osmotic membrane bioreactor (OMBR)–MFCs system, the membrane fouling in the OMBR was alleviated by the MFC treatment, and the electricity generation in the MFC was enhanced due to increased solution conductivity after the OMBR treatment. FO-based processes have also been studied as pre-treatment before BES shown in Figure 5. For example, an FO unit containing anaerobic acidification converted complex organic contaminants into short-chain fatty acids and alcohols, as well as concentrated wastewater [23], which was then treated in an MFC for electricity generation. In addition, the MDC-FO system can be applied for desalination, and the effluent salinity from MDC-FO system is lower than the maximum contaminant levels of the National Secondary Drinking Water Regulations. Compared to the integration of MDC and RO, the MDC-FO system might have lower energy consumption and lower membrane fouling propensity [24].

◾
**Enhance power generation performance**


Yao et al. [25] used the FO membrane as separator to build a new OsMFC. They found that OsMFC generated more electricity than MFC in batch operation and continuous operation. According to the polarization curve, the maximum power density of OsMFC is 4.74 W/m^3^, 36% higher than that of MFC with CEM, when 58 g L^−1^ NaCl is used for cathode liquid and aeration is used. The catholyte used 35 g/L NaCl, and the power density of the air cathode OsMFC is 8% and 87% higher than that of the MFC with AEM and CEM, respectively. Generally, the performance of MFC was evaluated by open circuit voltage and internal loss, including ohmic loss, activation loss, microbial metabolism loss, and concentration loss [26]. When the reactor configuration and electrolyte were the same, the open circuit voltages of OsMFC and MFC were not the same. Therefore, the main contribution to the improvement of OsMFC power generation capacity was the reduction of internal losses, such as low membrane internal resistance, low ion penetration resistance, and low pH gradient of cathode and anode solutions [27,28].

Zhao et al. [29] found that the membrane internal resistance of OsMFC is smaller than that of the MFC system and predicted that high water flux would reduce the internal resistance of the system after accurately simulating the experimental results with mathematical models. The air cathode OsMFC has a very low internal resistance, which is only 54 Ω. The resistance of ions passing through the FO membrane was 9 Ω, which is smaller than that passing through the AEM and CEM. This may be due to the existence of water flux, which accelerated the transmission speed of ions. After 10 h of operation, the pH of cathode solution of OsMFC is 9.76 and that of MFC is 10.90. This is because the rapid transport of protons in OsMFC buffers the continuously increasing pH of cathode solution, reduces the pH of cathode solution, and reduces the over voltage. At the same time, the water in the anode solution flows into the cathode solution through the forward osmosis membrane, leading to the concentration of the anode solution, which increases the conductivity of the anode solution, thereby reducing the internal resistance of OsMFC [30].

◾
**Recover high-quality water resources**


Compared with the ion exchange membrane, the FO membrane has a very high water permeability coefficient [31]. When the salinity of the catholyte is very high, high-quality water can move from the wastewater end, i.e., the anode chamber, to the cathode chamber through the FO membrane. For example, using 116 g L^−1^ NaCl solution as the catholyte of OsMFC can produce a water flux of 3.94 ± 0.22 LMH, and there is no water flux in MFC under the same experimental conditions. The cathode liquid after drawing water is purified by reverse osmosis, electrodialysis, or a desalination tank to remove the absorbent to achieve the purpose of water resource recovery [32]. In this regard, the cathode liquid of OsMFC acts as a catalyst for water purification and extraction. The water flux will also cause dilution of catholyte.

After the OsMFC run for 10–12 h, the conductivity of the catholyte decreased by 8% to 35%, which meant that this system was also a desalination system. Based on this discovery, Tiraferri et al. [33] proposed to build a forward osmosis microbial desalination cells (OsMDC). After three days of water dilution and salt removal, the conductivity of simulated seawater decreased by 60%. AEM in the traditional desalination tank allows chloride ions to pass through the FO membrane in OsMDC, which can intercept chloride ions and reduce the damage caused by the accumulation of chloride ions to microorganisms. It should be noted that the dilution of cathode solution and the concentration of anode solution caused by water molecule shuttle and RSF lead to the reduction of osmotic driving force [34]. In addition, with the increasingly serious membrane pollution, the water flux of OsMFC will gradually decrease. In addition, the concentration of anode solution improves the conductivity of anode solution, which is conducive to electron transfer and anode performance [35]. However, when the anode solution is concentrated to a certain extent, the salt concentration may inhibit the growth of anode microorganisms, thereby reducing the anode performance. This adverse effect can be reduced by increasing the anode solution circulation rate or increasing the desalination chamber. Periodic FO membrane cleaning, cathodic solution concentration and periodic replacement of anodic solution are necessary conditions for continuous water extraction. Generally, MFC includes the biological treatment process, which is slower than the FO process. The hydraulic retention time of the two processes is different, resulting in different treatment capacities. This imbalance reduces the treatment efficiency of the MFC anode for organic pollutants. In order to reduce the HRT gap between the two, proper coordination of MFC and FO processing capacity, such as increasing the size of anode cavity, can improve the performance of MFC system [36].

Compared with the traditional BES system, the most prominent feature of OsMFC is that it can extract high-quality water resources from wastewater through the embedded forward osmosis technology [37]. There is no obvious water infiltration flux on either side of the membrane in the traditional MFC system. Studies [38] have shown that the OsMFC system can recover more than 50% of the water resources from a variety of sewage, i.e., more than half of the water resources can be reused instead of being discharged directly by using the OsMFC technology to treat wastewater [39].

In OsMFC technology, a key factor for water recovery is the water transfer law under the influence of electric field and osmotic pressure. Since the membrane can be considered as a gel structure composed of cross-linked polyelectrolytes and forming water adsorption in the aqueous solution [40], the water content of the membrane, the concentration of salt solution on both sides of the membrane, temperature, and the density of fixed charges on the membrane surface have a greater impact on the water distribution and electroosmotic coefficient in the membrane. It has been reported that increasing the concentration of the salt solution of the extraction solution can increase the osmotic pressure on both sides of the cathode and anode, promote the migration of water to the anode, improve the water distribution in the membrane, and improve the battery performance. In addition to osmotic pressure driving on both sides of the membrane, another driving mode of water in the membrane is that it is carried by protons under the drag of electroosmosis and moves from anode to cathode [34]. The more protons cross the membrane, the greater the water flux that moves with protons from anode to cathode. The transmembrane water transfer phenomenon not only affects the size of the water flux in the forward osmosis process, but also the membrane impedance, because for any partition membrane, its ability to transmit protons is closely related to the water content of the membrane [41]. When the water content of the membrane is low, the conductivity of the electrolyte membrane is limited, while the water content of the membrane is closely related to the water transfer mechanism in the membrane. Therefore, by studying the water transfer phenomenon in OsMFC, the operating conditions of the battery can be optimized to ensure a higher and more stable output performance. Therefore, the in-depth study of water transfer phenomenon is of great significance for us to understand forward osmosis microbial fuel cells [42]. In addition, typical FO technology uses very high circulation speed to generate film side shear force to prevent pollution from accumulating on the film surface to reduce external concentration polarization. Generally, the solution circulation speed in OsMFC is smaller than that in typical FO system, and the cross flow speed on the membrane surface is 0.01~0.02 m s^−1^ vs FO 10~30 m s^−1^.High circulation speed cannot be used on the side of anode electrode with attached biofilm, otherwise the electrogenerating bacteria will fall off from the electrode. Therefore, the water resources recovered by OsMFC system will be higher than that of traditional FO system. The forward osmosis microbial fuel cell technology is a new sewage treatment and energy recovery technology that can effectively treat pollutants, purify water quality, and convert pollutants into electricity [43]. The organic matter is oxidized under the action of anode microorganism, releasing protons and electrons. The electrons first arrive at the anode through a series of transfers, and then finally arrive at the cathode through the external circuit to complete the reduction reaction. At the same time, the protons generated simultaneously with the electrons arrive at the cathode through the membrane and electrolyte to complete the current transfer, realizing the process of converting the chemical energy in the organic matter into electrical energy [44].

### 2.2. Development of Forward Osmosis Technology and Microbial Fuel Cell Technology

Forward osmosis microbial fuel cells (OsMFCs) represent a new type of microbial fuel cells formed by the combination of forward osmosis technology and microbial fuel cells. By combining the advantages of microbial fuel cell and forward osmosis technology, OsMFCs can use FO membrane to treat, concentrate and prevent the penetration of solute ions in the feed solution, i.e., MFCs anode wastewater, while generating electricity, and extract water from the anode electrolyte to the electrolyte through osmotic pressure [45]. Compared with conventional MFCs, OsMFCs can use sodium chloride solution or simulated seawater as cathode electrolyte to generate more electricity in intermittent mode and continuous mode as shown in Table 1. The improvement of its performance is due to the fact that the internal resistance of OsMFCs is lower than that of traditional MFCs [46]. Verma et al. [47] proposed a mathematical model of OsMFCs, which predicted that the internal resistance would decrease with the increase of osmotic pressure and water flux, and that the electricity generation would increase synchronously, thus confirming the importance of membrane resistance. Moreover, they believed that the lower membrane resistance in OsMFCs was related to the lower transmembrane pH gradient. This is because the water flux promotes the proton transfer. Compared with CEM, the combination of FO membrane and MFC technology can slow down the accumulation of cathodic pH, which has a good application prospect in the field of wastewater treatment. Previous studies have shown that FO membrane as separator of MFC has higher power generation performance than traditional MFC, which may be due to water flux accelerating proton transfer, low internal resistance, or reverse diffusion of salt to improve anode conductivity. However, the research on the mechanism of improving the power generation capacity is not clear, nor has a relatively consistent view been formed. At the same time, previous research has focused on the comparative analysis of electrochemical indicators, such as the power generation effect, while the research on the internal characteristics of the membrane, e.g., the impact of water flux on the membrane impedance of the forward osmosis membrane, as well as the distribution of salt concentration in the membrane and its relationship with the membrane impedance, is less or even blank [48]. Meanwhile, the main factor affecting its operation effect is the internal resistance power loss caused by high internal resistance, which mainly includes ohmic internal resistance loss, activation internal resistance loss and concentration difference loss. Research shows that using FO membrane to replace CEM or PEM can affect the membrane impedance in ohmic internal resistance [49]. However, given the characteristics of OsMFC, how the water flux affects its power generation capacity is worth further discussion. Previous research reports have confirmed that OsMFC, due to the generation of water flux, promotes the ion transport between the cathode and anode chambers, which indicates that FO membrane, as the separation material of MFC, has a lower blocking effect than CEM and AEM membranes as shown in Table 2. At the same time, the water flux can also promote the transfer of protons, easing the decrease of anode pH and the increase of cathode pH. The research results show that OsMFC can stabilize the system pH, thereby reducing the system overvoltage. In addition, the absorbing solution in OsMFC is usually a salt solution with relatively high concentration, so it has lower solution impedance than the MFC system, which can reduce the ohmic impedance loss of the whole system [50]. Since the operating conditions of FO membrane are the existence of osmotic pressure difference and concentration gradient on both sides of the membrane, the research on the characteristics of the membrane operating under the concentration gradient is not comprehensive at present [51].

### 2.3. Challenges of Forward Osmosis Technology and Microbial Fuel Cell Technology

Although OsMFCs technology has greatly improved its power generation capacity, as a technology based on the FO principle, OsMFCs also present some inherent disadvantages of FO, the most important of which is that reverse salt flux is almost inevitable, and it is also one of the most challenging problems. The reverse salt flux occurs due to the concentration gradient on both sides of the FO membrane, resulting in the reverse transport of the extracted solute to the side of the feed solution [52]. In the FO process, the ideal FO membrane should have high water permeability and low solute permeability to achieve high water flux while reducing the reverse salt flux. In OsMFCs, although the reverse salt flux can reduce the resistance of the anode solution, the excessive accumulation of salt will affect microbial activity, causing microbial dehydration while polluting the feed solution water quality. On the other hand, the impedance of the forward osmosis membrane as the separation material of MFCs is not constant and will change with the concentration of solution salts [53]. Especially when the concentration of the feed solution and the draw solution on both sides of the membrane are quite different, the key factors affecting the membrane resistance have not been studied in depth, and the relationship between the concentration of the outer membrane solution, the concentration of the inner membrane solution, and the membrane impedance still needs to be further explored.

## 3. Conclusions and Prospects

Forward osmosis (FO) technology has been developed to treat wastewater. The water flow in FO flows naturally from medium with high water concentration to medium with low water concentration [54]. The water treatment process consists of a semi permeable membrane that allows water to pass through and expel solutes. FO has attracted much attention due to its excellent energy efficiency and salt discharge capacity, as well as its low scaling tendency and saltwater discharge. Therefore, the synergy of microbial fuel cells and FO can potentially eliminate the dependence on fossil fuels, as well as provide better waste management. One technology to achieve this result is the OsMFC [55]. It can potentially be used in many processes, such as wastewater treatment facilities, where clean water can be produced and extracted, and in water desalination facilities where salt can be removed from water and used for water reuse.

OsMFCs seems to be more effective than conventional MFC in terms of energy generation and water extraction, due to the presence of FO membrane in OsMFC. It also leads to more power generation than conventional MFC and provides an opportunity to extract water through the anode chamber. Because of the many positive characteristics of OsMFC, they can be applied to many processes in practice. However, the FO membrane fouling remains a major challenge for these internal configurations, as it is difficult to apply in situ membrane cleaning. All the above problems related to OsMFC will eventually lead to operation in a short time [56]. This is why OsMFCs long-term continuous operation has not been well studied in previous studies. Based on these facts, more research is needed to better understand the combination of MFC and FO. In general, the research of OsMFCs is still in its infancy, but the huge prospect of MFC and FO as separate technologies in resource recovery and progress will accelerate the development of OsMFCs technology. More efforts must be invested to identify application areas, understand energy issues, alleviate membrane pollution, and expand OsMFCs to the transition stage.

## Figures and Tables

**Figure 1 membranes-12-01254-f001:**
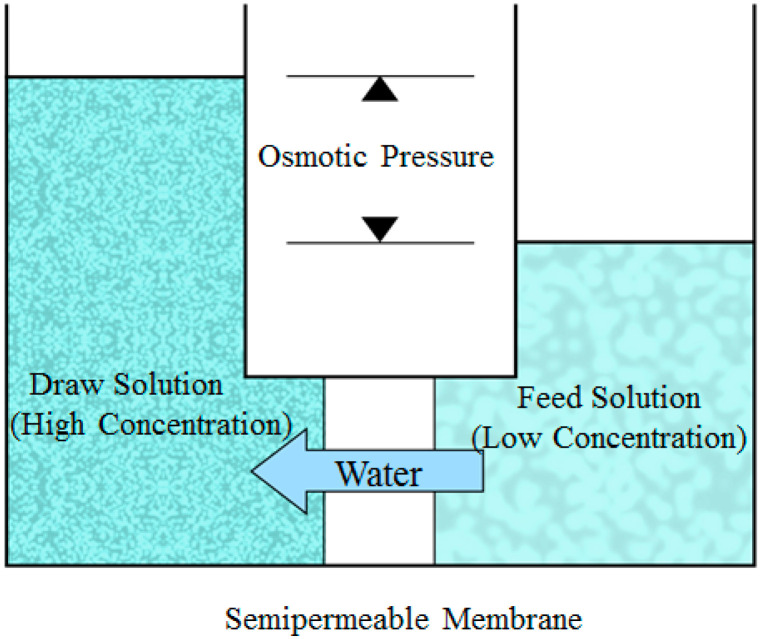
Schematic Diagram of Forward Osmosis Process.

**Figure 2 membranes-12-01254-f002:**
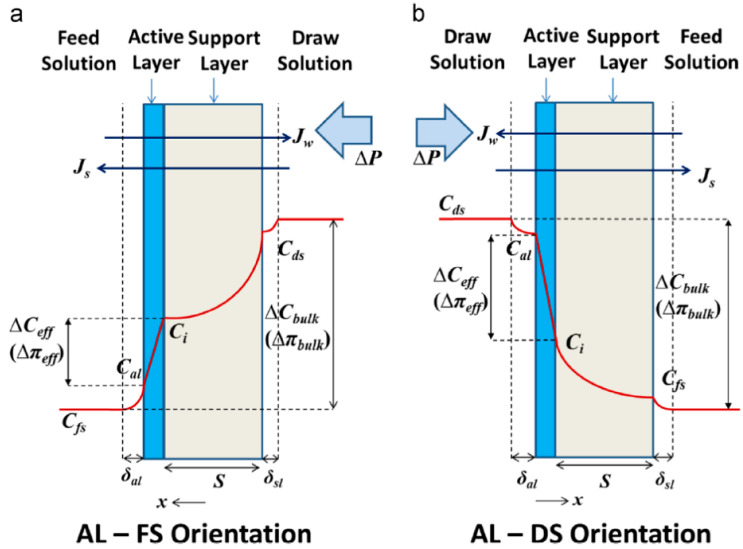
Schematic Diagram of Concentration Polarization in AL-FS and AL-DS Modes.

**Figure 3 membranes-12-01254-f003:**
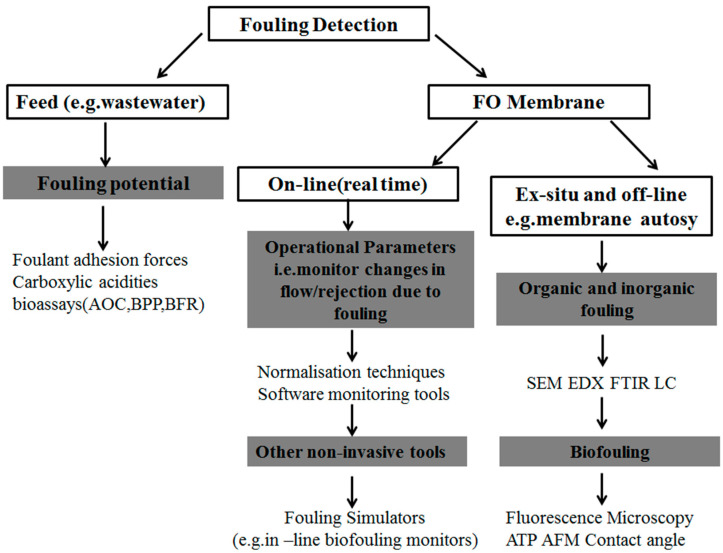
Methods used in literature to detect the fouling potential of the feed or monitor/analyse fouling on the FO membrane.

**Figure 4 membranes-12-01254-f004:**
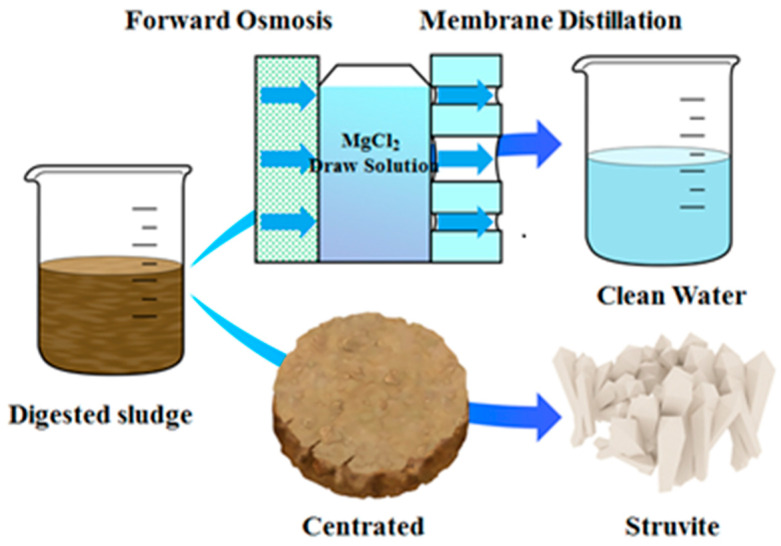
Schematic diagram of MgCl_2_ as draw solution to drive FO process and recover nitrogen and phosphorus from sludge digestion liquid.

**Figure 5 membranes-12-01254-f005:**
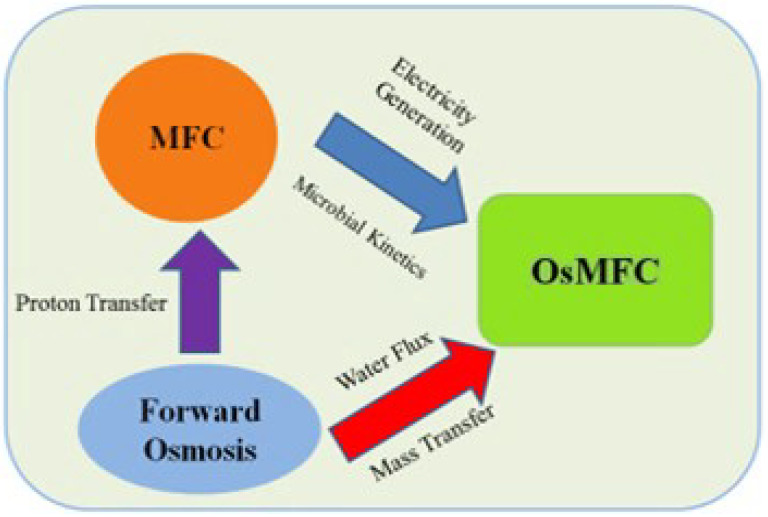
Coupling advantages of forward osmosis technology and microbial fuel cell.

**Table 1 membranes-12-01254-t001:** Summary of the OsBES studies for water recovery and energy recovery.

System	Membrane Type	Anolyte Substrate	Catholyte Solute	Catholyte Concentration	COD Removal (%)	Water Flux (LMH)	Maximum Power Density (W/m^3^)	References
OsMFC	CTA (Flat sheet)	Acetate	NaCl	20–116 g/L	82.5%	1.2–2.8	28.2	[18]
OsMFC	TFC-1	Acetate	NaCl	1 M	75.5%	1.82	13.6	[19]
OsMFC	TFC-2	Acetate	CaCl_2_	1 M	78.3%	0.56	7.3	[19]
OsMFC	TFC-3	Acetate	PBS buffer	1 M	65.6%	2.42	5.5	[19]
OsMFC	TFC-4	Acetate	Glucose	1 M	52.1%	1.82	3.7	[19]
OsMDC	CTA (Hydrowell filter)	Acetate	NaCl	5–20 g/L	-	0.29–0.69	-	[20]
OsMFC	CTA + polypyrrole	Glucose	NaCl	2 M	89.8%	1.1	27.8	[21]
OsMFC	Chitosan + PAAc	Glucose	NaCl	35 g/L	74.8%	18.4–34.4	24.5	[22]
OsMFC	TFC	Glucose	NaCl	58.5 g/L	85%	3.25	16.5	[23]
OsMEC	CTA	Acetate	PBS buffer	24 g/L	-	-	-	[24]
MEC-FO	TFC + Disulphonate	Glucose	NH_4_HCO_3_	0.8 M	60.6%	3.0		[24]
AAFO-MFC	TFC (DS-11-AG)	Acetate	NaCl	5 M	71.2%	2.33–5.62	4.38	[25]
MEC-PRO	CTA	Acetate	NaCl	0.1–2.0 M	80%	0.5–1		[25]
MFC-OMBR	TFC + Polydopamine	Acetate	NaCl	0.5	-	2.0–12	11.5	[26]
FO-MDC	CTA	Acetate	NaCl	35 g/L	70.6	0.64–0.99		[26]
OsMFC	CTA + Anthraquinone	Primary effluent	NaCl	1 M	74.8%	1.11–1.49	4.5	[27]
OsMFC	CTA	Domestic Wastewater	NaCl	58.5 g/L	85–90%	2.93	0.48–0.52	[27]
OsMFC	CTA-Double Skinned	Acetate	Oil produced water	-	-	1.8–4.1	3.9	[28]
OsMFC	CTA-Hydrowell filte	Acetate	NaCl	3 g/L	65%	0–0.75	0.615	[28]

**Table 2 membranes-12-01254-t002:** Comparison of conventional anaerobic digestion technology with microbial fuel cells technology and forward Osmosis Microbial fuel cells technology [45,46,47,48].

Items	Anaerobic Digestion Technology	Microbial Fuel Cells Technology	Osmosis Microbial Fuel Cells Technology
Configuration	Upflow Anaerobic Sludge Blanket (UASB) reactor.	Single/Two chamber.	Two chamber.
Biocatalyst	A complex “food chain” type microbial consortium catalyzes the process.	The microbial catalysts can be an axenic culture or a mixed culture.	Directly inoculated from other MFC reactors that have been domesticated and matured.
Power input	Can application for both high and low concentration COD biomass at temperatures about 30 °C.	Can be utilized rather low strength influents containing glucose, sucrose or acetate at temperatures below 30 °C.	-
Power output types	✓ 1/3 of the biogas produced is converted with a high energy level; ✓ 2/3 with a low energy level, which can be used to heat the digester.	Convert energy available in biomass directly to electricity.	✓ Recovery of electric energy and other energy from organic wastewater; ✓ Extracting high-quality water resources from wastewater; ✓ Removal and recovery of nutrients.
Power output units	The power density obtained is about 400 W/m^3^ when the technology is applied to treat about 5 to 25 kg of COD per m^3^ of the reactor per day.	The average power density of MFCs is about 40 W/m^3^.	Recently, stacked configurations of OsMFCs have reached power densities of 250 W/m^3^.
Advantages	✓ Removal of higher organic loading; ✓ Low sludge production and high pathogen removal; ✓ Low energy consumption.	Less excess activated sludge; Intensive to operation environment; Widespread application in location with insufficient electrical infrastructures.	In addition to retaining the advantages of MFC, the low membrane resistance enhances the power generation performance of MFC.
Disadvantages	✓ Difficult to store biogas; ✓ High cost to remove H_2_S.	✓ Limited effectivity of the open-air cathodes; ✓ High cost of electrode materials and proton exchange membrane.	Inhibition of Reverse Salt Flux and Recycling of draw solution.

## Data Availability

The data presented in this study are available on request from the corresponding author.

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
