# Peer review of "Forward Osmosis Technology and Its Application on Microbial Fuel Cells: A Review"

_membranes, 2022, doi:10.3390/membranes12121254_

Round 1

Reviewer 1 Report

In general, the article is tackling a very important and interesting topic, discussing the principle, development process and application of forward osmosis technology in wastewater treatment and energy recovery are reviewed in detail and systematically. Especially, the coupling with microbial fuel cell promotes the sustainable development of this technology. In order to better explain the development and application of this technology.Here are some suggestions:

1)    The authors should add to the introduction an explicit description of the research situation in the field of microbial fuel cell, including progress made and problems, and how these problems are solved by using forward osmosis technology

2)    As a unique feature of forward osmosis technology, how does water flux promote power generation in forward osmosis microbial fuel cells? Is there any relevant literature report

3)    The sentence in lines 210-212 requires clarification and reference to the source.

4)    In this paper, you mentioned that the FO membrane is different from the traditional membrane in that it can continue to generate electricity after membrane pollution. How about the effect of electricity generation compared with the new membrane.

5)    Compared with traditional water treatment technology, what are the advantages of forward osmosis technology, coupling with microbial fuel cells and what are the new changes

Reviewer 2 Report

The paperwork is attractive, and well done in general aspects, and I congratulate the authors. However, the manuscript presents some deficiencies, which require their review, necessary for it to be published. 

1.     What is the main question addressed by the research? Experimental R&D was missing.

2.     The manuscript should be revised carefully to avoid grammatical and typo errors and to improve the language.

3.     Authors should state the complete form of each abbreviation at its first appearance.

4.     Some of the recent references are cited in the suitable text (Chemical Engineering Journal, 2023, 451, 138767; Microbiological research 2022,  265: 127216; Energy Conversion and Management: X, 2022, 13: 100160;  International Journal of Hydrogen Energy. 2021,46(4):3220; Systems Microbiology and Biomanufacturing. 2022, 2, 67; Water 2022, 94, e10802, and so on.)

5.     List the different materials in a tabular form in the R&D results topic.

6.     Author should add challenges and prospects for future work.
